# RAGE Inhibitors in Neurodegenerative Diseases

**DOI:** 10.3390/biomedicines11041131

**Published:** 2023-04-09

**Authors:** V. Prakash Reddy, Puspa Aryal, Pallavi Soni

**Affiliations:** Department of Chemistry, Missouri University of Science and Technology, Rolla, MO 65409, USA

**Keywords:** RAGE, AGEs, Alzheimer’s disease, traumatic brain injury, RAGE antagonists, soluble RAGE, oxidative stress, glycation, cytokines

## Abstract

Nonenzymatic reactions of reducing sugars with primary amino groups of amino acids, proteins, and nucleic acids, followed by oxidative degradations would lead to the formation of advanced glycation endproducts (AGEs). The AGEs exert multifactorial effects on cell damage leading to the onset of neurological disorders. The interaction of AGEs with the receptors for advanced glycation endproducts (RAGE) contribute to the activation of intracellular signaling and the expression of the pro-inflammatory transcription factors and various inflammatory cytokines. This inflammatory signaling cascade is associated with various neurological diseases, including Alzheimer’s disease (AD), secondary effects of traumatic brain injury (TBI), amyotrophic lateral sclerosis (ALS), and diabetic neuropathy, and other AGE-related diseases, including diabetes and atherosclerosis. Furthermore, the imbalance of gut microbiota and intestinal inflammation are also associated with endothelial dysfunction, disrupted blood–brain barrier (BBB) and thereby the onset and progression of AD and other neurological diseases. AGEs and RAGE play an important role in altering the gut microbiota composition and thereby increase the gut permeability and affect the modulation of the immune-related cytokines. The inhibition of the AGE–RAGE interactions, through small molecule-based therapeutics, prevents the inflammatory cascade of events associated with AGE–RAGE interactions, and thereby attenuates the disease progression. Some of the RAGE antagonists, such as Azeliragon, are currently in clinical development for treating neurological diseases, including AD, although currently there have been no FDA-approved therapeutics based on the RAGE antagonists. This review outlines the AGE–RAGE interactions as a leading cause of the onset of neurological diseases and the current efforts on developing therapeutics for neurological diseases based on the RAGE antagonists.

## 1. Introduction

The non-enzymatic reaction of proteins, aminoglycosides and amino-terminal lipids with reducing sugars, such as D-glucose, followed by a sequence of Amadori rearrangements and oxidative modifications would give rise to what are commonly called Advanced Glycation Endproducts (AGEs) [1,2]. The AGEs have diverse structures and only a limited number of AGEs have been isolated and structurally characterized. AGEs may consist of insoluble, aggregated proteins, as well as small molecule compounds. Of the many AGEs that are isolated and characterized, pentosidine, glucosepane, Argpyrimidine and N^ε^-(carboxymethyl)lysine (CML) are relatively more abundantly formed in these nonenzymatic Maillard reactions. The latter AGEs are small molecule compounds, which may also be formed through the proteolytic degradation of the protein-crosslinked or protein-modified AGEs (Figure 1). The imbalance in the formation and destruction of AGEs under physiological conditions, especially under elevated oxidative stress conditions, leads to the build-up of excessive amounts of AGEs and thereby leads to the AGE-related disease progression. The exogenously ingested AGEs also contribute to the onset and progression of disease. Although under normal conditions, exogenous as well as endogenously formed AGEs are metabolized to nontoxic metabolites that are excreted from the body, under enhanced oxidative stress conditions, the formation of AGEs overwhelms the detoxification mechanisms and contributes to various neurodegenerative diseases, including Alzheimer’s disease (AD), traumatic brain injury (TBI) and amyotrophic lateral sclerosis (ALS) [3,4,5,6,7].

Although pathogenesis due to AGEs is multifactorial, recently it has been recognized that a key factor in the pathogenesis of AGEs is through their binding to receptors for Advanced Glycation Endproducts (RAGE), which results in a deleterious activation of a cascade of signaling events, and the formation of pro-inflammatory cytokines that triggers further oxidative stress and an excessive build-up of AGEs and RAGE. That is, the higher concentration of AGEs would lead to the increased expression of RAGE and this vicious cycle leads to the onset of neurodegenerative diseases, including AD, ALS, progression of the secondary effects of TBI, diabetes, atherosclerosis, rheumatoid arthritis, and cancers [8,9,10,11,12,13,14]. Other endogenous ligands, including high mobility group box1 (MGB1) proteins, also contribute to the RAGE-induced inflammatory responses (through intracellular signaling activation of the TLR4/NF-κB/interleukin pathway), as demonstrated in the TBI animal models [15]. The AGE–RAGE-mediated neuronal damage has received renewed attention as a causative factor in neurological diseases in recent years. The neuronal damage caused by AGE–RAGE interactions, followed by the activation of the inflammatory signaling cascade, is associated with the onset of various neurological disorders, including AD, TBI, ALS, and diabetic neuropathy and nephropathy. Furthermore, the AGE–RAGE interactions are also implicated in a variety of other inflammatory-associated diseases, including SARS-CoV-2 infections, diabetes, cancer, cardiovascular disease, inflammatory bowel disease, and bronchopulmonary dysplasia [1,16,17,18,19,20,21,22]. 

The increasing understanding of the molecular mechanisms leading to the AGE–RAGE-mediated neurological disorders has broadly impacted the design of RAGE inhibitors and antagonists for the treatment of neurological disorders and other RAGE-related diseases. Toward integrating the molecular mechanisms and recent progress in the design of RAGE inhibitors, in this perspective review we will outline the chemistry and biology of AGE–RAGE interactions and emerging small molecule-based therapeutical candidates for treating neurological disorders. We have used the SciFinder-n^®^ search using various key words related to AGE–RAGE interactions, AGE inhibitors and AGE antagonists and neurological disorders. We have emphasized on the most recent articles and also provided some of the earlier references that have fundamentally advanced the area of AGE–RAGE interactions. 

The nonenzymatic reaction of reducing carbohydrates (e.g., D-glucose, D-ribose) with the primary amino groups of proteins (or nucleic acids and lipids) reversibly forms the Schiff bases (aldehyde imines). Under physiological conditions, these reactions are relatively slow and are reversible reactions. However, under increased oxidative stress conditions, these Schiff bases undergo a further sequence of Amadori rearrangements, followed by oxidative degradation to form the AGEs (Figure 1). These reactions can also be achieved under laboratory conditions and various AGEs can be synthesized and compared with the AGEs formed under physiological conditions. 

The AGEs are soluble or insoluble protein aggregates and may be fluorescent or non-fluorescent [23]. AGEs exert their toxicity through various mechanisms, including protein crosslinking, AGE–RAGE interactions (and thereby inducing formation of the pro-inflammatory cytokines) and interactions with metal ions to release reactive oxygen species (ROS) that, when present in excessive amounts, would lead to damage to nucleic acids, proteins and lipids. Protein crosslinked AGEs include pentosidine and glucosepane, the lysine and arginine crosslinks. Proteolysis of these crosslinks under normal physiological conditions releases the soluble forms of pentosidine and glucosepane from the crosslinked proteins. Protein modifications that are non-crosslinked, such as Argpyrimidine, CEL (N^ε^-carboxyethyllysine) and CML (N^ε^-carboxymethyllysine), also disrupt normal protein functions and may lead to pathogenesis through their interactions with the extracellular domain of the RAGE. There are numerous AGEs that are not structurally characterized because of their low abundance in tissues. Some of the well characterized AGEs include pyrroline, methylglyoxal–lysine dimer (MOLD), hydroimidazolones (methylglyoxal derived AGEs) and glyoxal–lysine dimer (GOLD). AGEs formed in the extracellular regions of the neuronal cells also bind to the amyloid-β (Aβ) plaques contributing to the production of reactive oxygen species (ROS), and thereby exacerbate oxidative stress. Furthermore, elevated levels of AGEs result in the expression of a higher concentration of RAGE (receptors for advanced glycation endproducts), which through binding to AGEs and other RAGE ligands activate the downstream (intracellular) signaling pathways, resulting in the expression of pro-inflammatory nuclear factor kappa beta (NF-κβ) and inflammatory cytokines, such as interleukin-6 (IL-6) [1,24,25].

In addition to the endogenously formed AGEs in the human body, dietary AGEs (dAGEs), AGEs that are present in modern diets, especially those derived from animal-derived foods, may also be contributing to the emerging epidemics of diabetes and cardiovascular disease and neurodegenerative diseases [26,27]. The animal derived foods, in comparison to those based on plant-based foods, are rich in fats and proteins and have abundant AGEs, which upon cooking at excessively high temperatures form secondary AGEs [26]. These dAGEs, similarly to the AGEs, may exert their toxicity effect either through interactions with RAGE, followed by an inflammatory cascade of events, or through forming complexes or crosslinks with various proteins, thereby inactivating proteins. Thus, the dAGEs augment the endogenously formed AGEs and collectively contribute to the onset of AGE-related diseases, including AD [27,28]. In this context, it is to be noted that dietary polyphenols may potentially prevent the neurological diseases via modulation of the AGE–RAGE axis and the microbiota–gut–brain axis [29].

The AGE-crosslinked proteins would lead to various pathological effects, depending on the location of the AGEs. For example, the AGE-modified amyloid-β peptides (Aβ_1–42_, Aβ_1–40_) are abundant in the intercellular spaces of neuronal cells and they ultimately are the causative factors in neuronal disintegration and thereby the onset of neurodegenerative diseases, including AD, Parkinson’s disease and ALS [30,31,32,33,34,35,36,37]. Furthermore, the AGEs are abundantly formed in the vascular tissue of diabetic and atherosclerosis cases [38,39,40]. AGE–RAGE interactions trigger downstream signaling and the exacerbation of oxidative stress and inflammation, leading to the onset of various lung cancers, pulmonary fibrosis, cystic fibrosis and pneumonia [41,42].

## 2. Receptors for Advanced Glycation Endproducts (RAGE)

Receptors for Advanced Glycation Endproducts (RAGE) and their binding to the AGEs were first demonstrated by Schmidt and coworkers in 1992 [43]. Using radio-labeled ^125^I-AGE albumin as the AGE substrates, Schmidt and coworkers demonstrated the binding of AGEs to the RGE-expressing endothelial cells. In more recent studies, RAGE, in addition to binding to AGEs, were shown to bind to multiple other ligands, including amyloid-β (Aβ), S100/calgranulins and high-mobility group box1 proteins (HMGB1; amphoterin) and thereby elicit a variety of signaling pathways that lead to the pro-inflammatory cytokines formation [44,45,46,47]. While HMGB1 acts as a DNA chaperone, repairing the DNA damage in the nucleus, in the extracellular region it elicits inflammatory responses through its binding to the extracellular domain of RAGE [22]. Anti-HMGB1 monoclinal antibody (Anti-HMGB1 mAb) was shown to ameliorate the inflammatory effects in TBI [15]. Glycyrrhizic acid, a saponin extracted from liquorice (licorice) roots, similarly inhibits RAGE acting as an antagonist of the HMGB1 (Figure 2) [35,48,49].

The AGE–RAGE interactions and binding of RAGE with various other ligands not only contributes to the exacerbation of oxidative stress but also to the over-expression of RAGEs themselves. The signaling pathways arising from AGE–RAGE interactions are implicated in a variety of pathological disorders, including neurodegeneration, cardiovascular diseases, cancer, diabetic neuropathy, diabetic retinopathy and diabetes [8,9,10,11,12,13,14,50]. Furthermore, in Parkinson’s Disease (PD) cases, microglial cell-mediated neuroinflammation and α-synuclein aggregation are exacerbated upon the binding of RAGE to receptors on α-synuclein fibrils on microglia [34]. 

AGE–RAGE interactions result in the upregulation of lysyl oxidase and endothelin-1 in human aortic endothelial cells and induce the upregulation of mitogen-active protein kinase (MAPK), thereby leading to the activation of the nuclear factor kappa B, NFκB and activator protein-1 (AP1). Thus, AGEs in concert with their binding to RAGE activate parallel ERK1/2-NF-κB and JNK-AP-1 signaling cascade sin human endothelial cells, affecting cell proliferation and endothelial integrity [51]. Disruption of the blood–brain barrier (BBB), consisting predominantly of endothelial cells, is one of the risk factors in the onset of AD and thus AGE–RAGE interactions have also impacted on the integrity of the BBB [52].

The AGEs bind to the ectodomain (i.e., extracellular domain) of the receptors for advanced glycation endproducts (RAGE), and this AGE–RAGE binding initiates a cascade of signal transduction events, including activation of nuclear transcription factor kappa beta (NFκB), and subsequent gene activation for the expression of the pro-inflammatory cytokines, such as interleukin-1β, interleiukin-6 and tumor necrosis factor-alpha (TNFα) (Figure 3). This, in turn, results in the exacerbation of oxidative stress and overproduction of AGEs and the vicious cycle of the enhanced AGE–RAGE interactions. AGE–RAGE binding also leads to activation of NADPH oxidase (NOX) and thereby expression of pro-inflammatory cytokines that are the sources overproduction of reactive oxygen species (ROS; for example, superoxide radical anion, hydroxyl radical), thereby exacerbating oxidative stress and elevating the levels of AGEs. 

### 2.1. Soluble RAGE

Soluble RAGE (sRAGE) are the C-terminal truncated RAGEs and because they lack the transmembrane domain, binding of AGEs to sRAGE will not result in the signaling events. Formation of sRAGE partly attenuates the AGE/RAGE induced cellular damage. sRAGE competes with RAGE in binding to AGEs and thus AGE/AGE interactions and the resulting signal transduction events are attenuated. Furthermore, sRAGE sequester the AGEs and thereby attenuate the AGE-induced toxicity effects. Administration of sRAGE to the RAGE^−/−^ mice was demonstrated to significantly reduce the infarct size in ischemic brain damage [14]. Overexpression of sRAGE in cardiomyocytes also resulted in improved cardiac function in ischemia/reperfusion (I/R) injury, presumably through their effect on the inhibition of mitochondrial depolarization and thereby cardiac apoptosis [53]. Administration of the recombinant versions of sRAGE was shown to suppress the activation of the cell surface RAGE [54]. sRAGE concentrations in plasma are relatively low in AD cases as compared to the control cases and may thus provide as a biomarker for the monitoring the progression of AD, although this biomarker assay is not reliable for the differentiation of AD from vascular dementia (VaD) and other types of dementia [55]. 

Metalloproteinase induced shedding of the full-length RAGE to the sRAGE has implications in the treatment of neurological disorders and diabetic complication; The metalloproteinases ADAM10 and MMP9 are involved in the proteolytic degradation of the cell surface RAGE to generate the sRAGE. G-protein-coupled receptors (GPCR), such as PACAP receptor (subtype PAC1) activate the expression of metalloproteinase through various mechanisms, including activation of Ca^2+^ signaling and MAP kinases, and thereby are involved in the shedding of the cell surface GPCR to the sRAGE [54]. This stimulation of RAGE shedding to sRAGE has impact in developing GPCR agonists as therapeutics for the AD and other AGE-related diseases, including diabetes and cardiovascular diseases.

Soluble RAGE (sRAGE), because of the lack of transmembrane domain, do not take part in the signaling cascade events. sRAGE are also formed through alternative splicing of the human RAGE mRNA. These sRAGE that are secreted from the spliced mRNA are also called as endogenous secretary RAGE (esRAGE). The esRAGE are abundantly expressed in the human vascular endothelial cells and pericytes and were shown to have cytoprotective against AGEs, and attenuate the diabetic vascular injury [56]. The esRAGE have the same sequence as that of the sRAGE and have similar roles in the trapping of the AGEs, thereby attenuating the toxicity effects of the AGEs. esRAGE, as the decoy ligands for AGEs, regulate the endothelial permeability, and it was shown that the esRAGE-expressed oncolytic herpes simplex virus 1 (osHV), when released into the endothelial cells, enhanced the therapeutic efficacy in glioblastoma-bearing mice [57]. The sRAGE and esRAGE, because they lack the transmembrane region, act as decoy receptors for AGEs and various other ligands to the RAGE, and thereby attenuate the toxic effects of AGEs [58]. 

RAGE-suppression results in the attenuation of ischemia-reperfusion injury in animal models, showing that the RAGE mediates the ischemic brain damage [14,59]. Although accumulation of AGEs in individuals is a normal phenomenon, excessive accumulation of AGEs results in various deleterious effects on cell metabolism, neuronal apoptosis, and induction of the tau-protein kinases that are involved in the formation of amyloid-β (Aβ) plaques and neurofibrillary tangles (NFT) in the brains of AD cases [1,24,60]. Therapeutic approaches using RAGE antagonists, AGE inhibitors, and recombinantly produced soluble RAGE (sRAGE), therefore may prove to be successful in the treatment of AD, diabetes, and other AGE-related pathologies [60].

Among other neuroprotective polyphenol antioxidants and polyphenolic metabolites, urolithin A was shown to attenuate the oxidative stress, downregulate inflammatory cytokines and inhibit NF-κβ activation, and was thus proposed as a potential therapeutic candidate, for treating inflammatory diseases, including neurodegenerative diseases [61,62,63,64]. Some of the chemically synthesized analogs of urolithins inhibit the binding of the bovine-serum albumin derived AGEs to the soluble RAGEs (AGE2-BSA/sRAGE) and their RAGE antagonist characteristics are comparable to that of the RAGE antagonists Azeliragon and FPS-ZM1, which are at various phases of clinical trials [65,66].

Plant extracts derived from *Artemisia herbalba* were demonstrated to attenuate the production of pro-inflammatory cytokines, such as TNF-α and interleukin-6 (IL-6), which are formed through AGE–RAGE signaling pathways [67]. Polyphenols from tea also regulate the RAGE expression and thereby attenuate the MAPK and TGF-β signaling pathways [68]. Polyphenols act as effective antioxidants by scavenging reactive oxygen species (ROS), and thereby attenuate the oxidative stress. The exacerbated oxidative stress would lead to the overproduction of AGEs and thereby increased AGE–RAGE interactions and the concomitant signal transduction pathways and the activation of the pro-inflammatory cytokines. Polyphenols regulate the microbiota–gut–brain axis and attenuate AGE–RAGE interactions and thereby the onset of neurodegenerative diseases and other pathological effects, as described above [29]. Ginkgolide-B, a constituent of Ginkgo biloba plants, also helps to decrease the levels of RAGE and thereby attenuate the oxidative stress and thereby neurological damage (Figure 4) [69].

### 2.2. Structural Aspects of Receptors for Advanced Glycation Endproducts (RAGE)

The mechanism of the AGE-mediated cellular damage is multifactorial and is not well established. Binding of the AGEs to the cell surface receptors, called receptors for advanced glycation endproducts (RAGE), initiate a cascade of intracellular signaling pathways, further exacerbating the disease condition. RAGE is expressed in a range of cells, including endothelial cells, vascular smooth muscle cells, and macrophages. N^ε^-(carboxymethyl)lysine (CML), an abundantly formed AGE (through lysine residue modification of proteins) is involved in the upregulation of RAGE [70]. The RAGE–CML interactions lead to the cytokine release, which further enhances oxidative stress, further contributing to the vicious cycle of the excessive formation of AGEs and RAGE. 

RAGE proteins are an immunoglobulin superfamily of receptors and constitute 35 kDa polypeptides. The cDNA-encoded RAGEs (human and bovine lung libraries), in in vitro studies were shown to bind to 125-Iabeled AGE-albumin with a dissociation constant (K_d_) in the range of 200 nM, and this binding is inhibited by RAGE antibodies, demonstrating that RAGE–AGE interactions can be modulated by RAGE inhibitors [38]. The N-terminus, extracellular region (ectodomain) of the RAGE consists of three unique immunoglobulin-like domains: Variable, V-domain (Ala23 to Val116), and constant domains, C1 (Pro124-Glu231) C2 (Glu236-Ile320). The X-ray crystallographic structure of the ecto-domain (PDB ID: 4LP5) shows that the VC1 unit has a large number of positively charged lysine (Lys39,43,37,44,52,62,110,123,140,174,169) and arginine residues (Arg29,216,57,98,116,48,77,104,178,221,228), and thus the VC1 region of the ectodomain binds largely to the negatively charged ligands, including AGEs, S100 family of proteins, high mobility group box1 proteins (HMGB1) and β-amyloid (Aβ) peptides. This ligand binding to the VC1 domain triggers intracellular signaling events, mediated by the 42-residue acellular RAGE domain [71,72]. The transmembrane region, residues 343–363 as well as the 42-residue cytoplasmic domains (residues 363–404) are probably unstructured [73]. X-ray crystallographic studies of RAGE revealed the multi-modal homo-dimerization and oligomerization through the V-domains, through the GxxxG motif, thereby facilitating ligand-specific complex formation and inducing intracellular signaling mechanisms through the activation of intracellular domain (Figure 5). The transmembrane region may also be involved in the homo-oligomerization reactions upon ligand binding, although the mechanisms of such oligomerization have not yet been elucidated [72]. 

The intracellular domain of RAGE binds to various proteins, such as ERK 1/2 (extracellular signal-regulated kinase 1 and 2), DIAPH1 (mammalian diaphanous 1) and TIRAP (Toll-interleukin 1 receptor adaptor protein), that when activated by the simultaneous binding of RAGE extracellular ligands, including AGEs, initiates downstream signaling events [71,74,75,76]. The latter signaling events are involved in the activation of NF-κβ, and thereby the expression of pro-inflammatory cytokines, such as IL-6, IL-9 and TNF-α. These pro-inflammatory cytokines, through exacerbated oxidative stress, not only activate the tau-phosphorylation and formation of Aβ peptides in AD but also upregulate RAGE and AGEs. 

## 3. RAGE and Gut Microbiota

The bacterial metabolites affect human health in many ways. Gastrointestinal microbiomes present in the gut help the digestion and metabolism of food and produce numerous metabolites, which would eventually impact on health and disease. The human intestinal microbiota metabolites are also correlated with the gastrointestinal diseases, including irritable bowel syndrome, inflammatory bowel disease and type-2 diabetes, and are also associated with the onset and progression of the neurological disorders, including AD, ALS and TBI. There is also extensive data showing the involvement of gut microbiota in the progression of cancer [77]. A fraction of the dAGEs that are not intestinally absorbed or excreted are metabolized by the gut microbiota, and in this process, they affect the gut microbiota composition, and induce insulin resistance in mice [78].

Dysbiosis of the gut microbiota, affected by AGEs and environmental factors, may impact on human disease, and there is increasing interest in modulating the gut microbiota to healthy levels toward restoring human health and developing translational therapeutics based on the gut microbiota [79].

Several studies have shown that the gut microbiome also protects the host by developing the immune system and preventing the colonization of the pathogens in the gut [80]. The administration of aged microbiota from aged rats (through fecal microbiota transplantation; FMT) to younger rats resulted in the reduced expression of brain-derived neurotrophic factor (BDNF), exacerbated oxidative stress and the increased expression of AGEs, RAGE and pro-inflammatory cytokines, showing a correlation with bacterial aging and the cognitive decline of the host [81]. These studies pointed to the correlation of cognitive decline with inflammation and oxidative stress. Furthermore, the communication system between the gut and the brain, called the microbiota–gut–brain axis, ensues the transmission of the neuronal, hormonal and immunological signals between the gut and the brain allowing the gut microbiota and its metabolites a pathway to access the brain [80]. This communication is bidirectional between the brain and the gut, and the gut microbiota dysbiosis disrupts the normal signaling pathways leading to neurological disorders, such as AD. Several recent studies show that AGEs and RAGE play an important role in changing gut microbiota composition and thereby increasing gut permeability and affecting the modulation of immune-related cytokines [1,82,83,84,85]. AD and other neurological disorders are associated with the imbalance of gut microbiota and intestinal inflammation. Although the mechanisms of the onset of neurological disorders as related to AGEs and RAGE is not yet clearly understood, reactive oxygen species (ROS) arising due to the AGE–RAGE interactions are importantly involved in the progression of neurodegenerative diseases, including AD. Polyphenols, due to their scavenging effect on ROS can effectively regulate the RAGE–AGE interactions and may play an important role in the prevention of neurological diseases [29]. Ginkgolide-B, derived from the Ginkgo biloba leaf extracts, was shown to be neuroprotective in AD model mice. GB significantly reduced the levels of RAGE and Bax (Bcl-2 like protein 4, an apoptosis regulator) in these AD model mice [69]. The neuroprotective effect of Ginkgolide-B was rationalized as partly due to its modulating effect on the gut dysbiosis [69]. 

### RAGE-Mediated Inflammatory Processes in Neurodegenerative Diseases

AGE–RAGE interactions and the accompanying inflammatory processes are the hallmarks for the onset of neurological diseases, including AD. In AD-like rat models, it was demonstrated that the methylglyoxal/RAGE/NOX02 pathway is activated in the hippocampus region [86]. These experiments suggested that the methylglyoxal, a precursor compound of AGEs, activates the membrane-bound NADPH oxidase (NOX), thereby leading to the expression of inflammatory markers, and the accompanying astroglial and microglial activation and synaptic dysfunction in animal models as well as in AD patients. Dietary methylglyoxal was also shown to induce AD through the increased expression of amyloid-β (Aβ) in AD models of *Caenorhabiditis elegans* (*C. elegans*) [87]. It was demonstrated that the use of a citrus flavonoid, nobiletin, attenuated the methylglyoxal-induced toxicity [87]. Other polyphenolic compounds, including ss-caryophyllene and carnosic acid, and camellia oil were also demonstrated to show anti-inflammatory activity and thus act as potential neuroprotective compounds [85,88,89]. The RAGE-induced activation of CD5L, an inflammatory protein, leads to the decreased brain volumes, while higher circulating soluble RAGE (s-RAGE), which functions as a decoy receptor for RAGE, was associated with increased brain volumes in AD models [90]. High mobility group box1 (HMGB1), a nuclear protein, interacts with RAGE, thereby inducing various inflammatory pathways that would lead to neurological diseases, including AD, and cardiovascular diseases [20]. In transgenic AD mice models, it was shown that curcumin improves the memory deficits via inhibition of the HMGB1-RAGE/TLR4-NF-kB signaling pathways [91]. 

Gut–microbiota dysbiosis results in exacerbated oxidative stress and increased levels of inflammatory cytokines and thereby resulting in age-related cognitive decline [81]. Gut microbiota is modulated by camellia oil (which contains oleic acid and other polyunsaturated fatty acids), and through its antioxidant effects exerts neuroprotective effects, as demonstrated in rat models of AD [85]. Polyphenolic antioxidants, in general, regulate the diversity of the gut microbiota and attenuate the formation of oxidative stress markers, including reactive oxygen species (ROS), and thereby prevent neurodegenerative diseases [29]. Through the attenuation of ROS, these polyphenolic compounds attenuate the AGEs formation and thereby suppress the AGE–RAGE interactions and delay or prevent the progression of neurodegenerative diseases. 

## 4. RAGE Antagonists 

RAGE is a multi-ligand receptor and the AGE-derived ligands include (N^ε^-carboxy-methyl)lysine (CML), (N^ε^-carboxyethyl)lysine (CEL), and methylglyoxal-derived hydroimidazolones (MG-H1). The CML and CEL-peptide/protein conjugates are bound to the V-domain of RAGE at the positively charged Lys52, Lys110 and Arg98 residues and hydrophobic residues [71,92]. MG-H1 has a high binding affinity to the V-domain of RAGE, in the nanomolar range. Unlike CML and CEL, MG-H1 does not necessarily need to be attached to the peptide/protein units for its binding to RAGE. It was demonstrated that the binding of the methylglyoxal-modified albumin to RAGE initiates signal transduction [92]. It binds to the positively charged residues of the V-domain and the relatively high affinity of this AGE ligand to RAGE was implicated it as the major AGE ligand to RAGE.

The broad significance of RAGE–AGE interactions has only recently been receiving significant attention toward developing therapeutics for neurodegenerative diseases as well as other AGE-related diseases, including diabetes, atherosclerosis and cancers [18,71,93,94,95,96,97,98,99]. RAGEs are involved in the onset of various pathological conditions, including cardiovascular diseases, diabetes, cancer and neurological disorders. RAGE antagonists may be either endogenous or exogenous (i.e., non-natural, often small molecule-based compounds) ligands that when bound to the RAGE attenuate the AGE–RAGE binding interactions and the downstream pro-inflammatory signaling events, and consequently prevent disease progression. The RAGE antagonists have received substantial interest in recent years toward developing therapeutic candidates. Recent reviews address the small molecule-based RAGE inhibitors, some of which, e.g., FPS-ZM1 (4-Chloro-*N*-cyclohexyl-*N*-(phenylmethyl)benzamide) and Azeliragon (3-[4-[2-butyl-1-[4-(4-chlorophenoxy)phenyl]-1*H*-imidazol-4-yl]phenoxy]-*N*,*N*-diethyl-1-propanamine), have gone onto clinical trials; although, currently none of the RAGE inhibitors/antagonists have been FDA-approved as therapeutics (Figure 6) [18,50,71].

Early studies showed that FPS-ZM1 selectively binds to RAGE and inhibits RAGE-mediated formation of Aβ_40_ and Aβ_42_, and inhibits β-secretase activity (an enzyme involved in the formation of Aβ peptides), and thereby attenuates Aβ-mediated brain damage [100]. Targeting the AGE–RAGE axis may also lead to the potential therapeutics for multiple AGE-related diseases, including diabetic nephropathy and cancer cell metastasis [101,102]. α-Synuclein binds to the positively charged residues of the RAGE at the V-domain through its acidic C-terminal residues, thereby inducing signaling pathways for the neuroinflammation. FPS-ZM1, through its selective binding at the positively charged residues at the V-domain, blocks the binding of α-synuclein fibrils with RAGE, thereby reducing the inflammatory response of microglia. Therefore, RAGE-inhibitors and antagonists, including FPS-ZM1 may emerge as therapeutics for Parkinson’s disease [34]. The ischemia-reperfusion-mediated hippocampal damage in animal models is attenuated by the FPS-ZM1 at moderate to high doses, but ineffective at relatively low doses, showing some of the therapeutic limitations of the latter potential drug candidate [103]. 

RAGE antagonist Azeliragon (TTP488) is currently in phase 3 clinical trials for treating AD. This drug candidate in phase 2b clinical trials showed decreased levels of Aβ plaques in the brain, with a concomitant increase in Aβ concentrations in plasma, reduced levels of inflammatory cytokines and slow cognitive decline [104]. The phase 3 clinical trials of Azeliragon are expected to demonstrate the slow cognitive decline in mild AD cases. Urolithin and its various structural analogs were demonstrated to inhibit AGE–BSA/sRAGE interactions, and their RAGE inhibition is comparable to that of Azeliragon [65].

AGE–RAGE interactions are substantially involved in diabetic neuropathy and retinopathy and RAGE antagonists would therefore provide as effective therapeutics for this disease [105]. The RAGE antagonist Azeliragon was shown to ameliorate the streptozotocin-induced diabetic neuropathy and its anti-nociceptive effect is comparable with that of the pregabalin, a potent analgesic drug [19]. 

Schmidt and coworkers have investigated a library of 58,000 small molecule competitive inhibitors of the cytoplasmic tail of RAGE (ct-RAGE), as RAGE-DIAPH1 antagonists, and identified 13 competitive inhibitors that in in vivo and in vitro experiments were effective in the blocking of AGE/RAGE-mediated signaling events. Through fluorescence titration experiments, these lead compounds were shown to exhibit a nanomolar affinity to the ct-RAGE (Figure 7). Twelve of these thirteen lead compounds thus generated exhibited significant suppression of CML-AGE mediated upregulation of IL-6 and all of the compounds significantly suppressed the CML-AGE mediated upregulation of TNF-α [76]. These small molecule antagonists thus complement the extracellular antagonists of RAGE toward the development of drug candidates for treating neurological disorders and other AGE-related diseases. It is interesting to note that some of these lead compounds for ct-RAGE have a structural similarity to that of FPS-ZM1, an extracellular RAGE antagonist. That is, they have benzamide as a common moiety. Quinoline-derived ct-RAGE inhibitors, RAGE203, RAGE208 and RAGE229 were selected as potential therapeutics for RAGE-mediated diabetic complications (Figure 7). In diabetic rat models, the latter compound RAGE229 reduced the plasma concentrations of TNF-α, IL-6 and CCL2/JE-MCP1 (plasma C–C motif chemokine ligand 2/JE (CCL2/JE) monocyte chemoattractant peptide-1 (JE/MCP1)) and mitigated diabetic complications [106]. In diabetic mice, the plasma levels of TNF-α are significantly elevated as compared to the non-diabetic mice and the compound RAGE229 lowered the levels of TNF-α and other inflammatory cytokines in both type1 and type 2 diabetes. These results show that the blocking of the ct-RAGE has a direct impact on the signaling pathway and prevents the formation of the inflammatory cytokines.

## 5. Conclusions and Outlook

AGE–RAGE interactions and the accompanying inflammatory cascade of events lead to the expression of the pro-inflammatory transcription factors and cytokines, and the ensuing exacerbated oxidative stress triggers the neuronal damage and the onset and progression of neurological diseases, including AD, PD, ALS, diabetic neuropathy and the secondary effects of TBI. The AGE–RAGE interactions and the accompanying signal transduction pathways also play an important role in altering the microbiota composition and this, in turn, affects the gut permeability and immune-related cytokines, and exacerbates the endothelial cell damage. Selective blocking of the AGE–RAGE interactions through small-molecule-based RAGE antagonists would attenuate the otherwise exacerbated oxidative stress and thereby also help to maintain microbiota homeostasis. The RAGE antagonists that would target AGE–RAGE interactions in the extracellular domain, including Azeliragon and FPS-ZM1, are currently in clinical development. Furthermore, the RAGE antagonists based on the cytoplasmic tail of the RAGE (ct-RAGE) block the signal transduction events of the extracellular domain of AGE–RAGE interactions. Recent studies demonstrated that the ct-RAGE antagonists attenuate the disease progression in diabetic animal models at nanomolar concentrations, and the latter small-molecule-based antagonists significantly attenuated the plasma concentrations of inflammatory cytokines. In this context, polyphenolic compounds, such as urolithins and ginkgolides, have a beneficial effect on modulating the gut–bacterial composition and therefore designing polyphenolic compounds that exhibit RAGE inhibitory effects would ideally lead to effective therapeutics for neurological disorders.

## Figures and Tables

**Figure 1 biomedicines-11-01131-f001:**
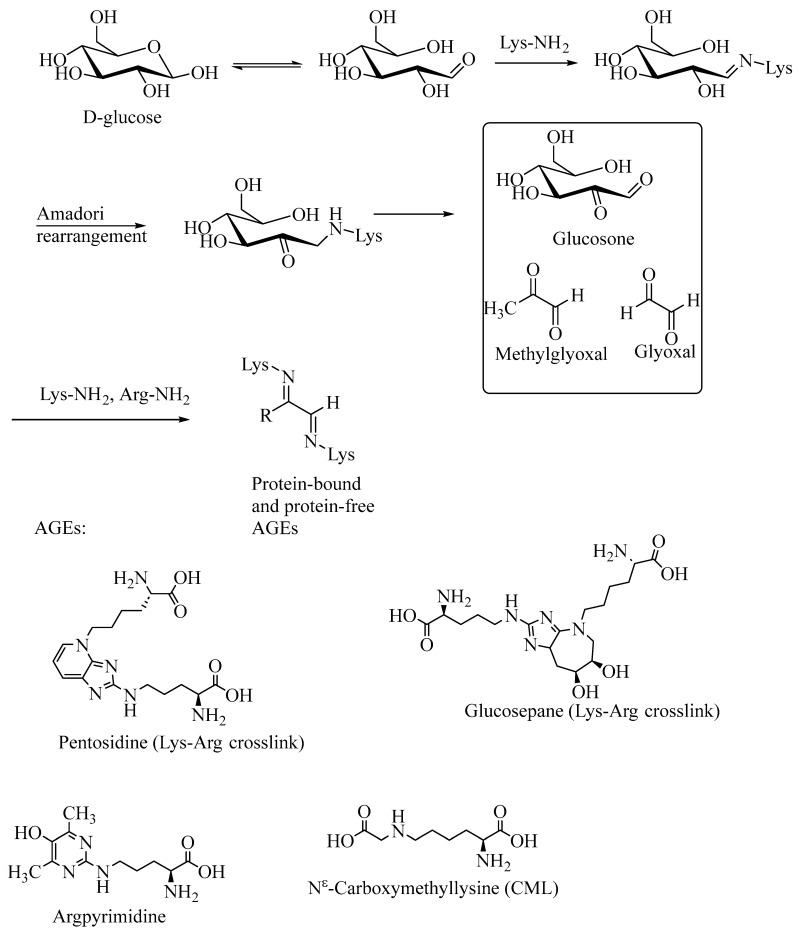
Nonenzymatic glycation of protein amino groups with reducing sugars to form the advanced glycation endproducts (AGEs).

**Figure 2 biomedicines-11-01131-f002:**
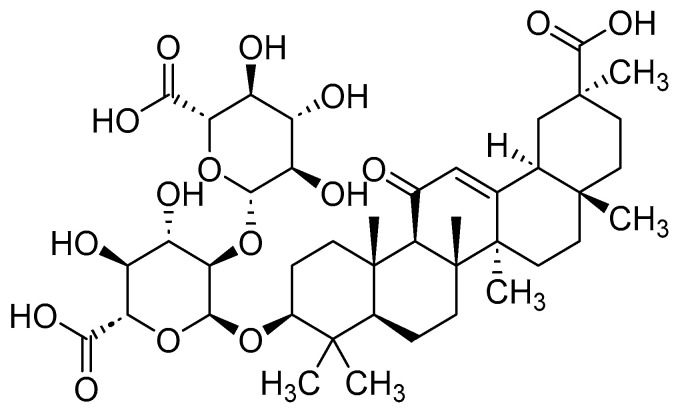
Structure of glycyrrhizic acid, a naturally occurring inhibitor of RAGE acting as HMGB1 antagonist.

**Figure 3 biomedicines-11-01131-f003:**
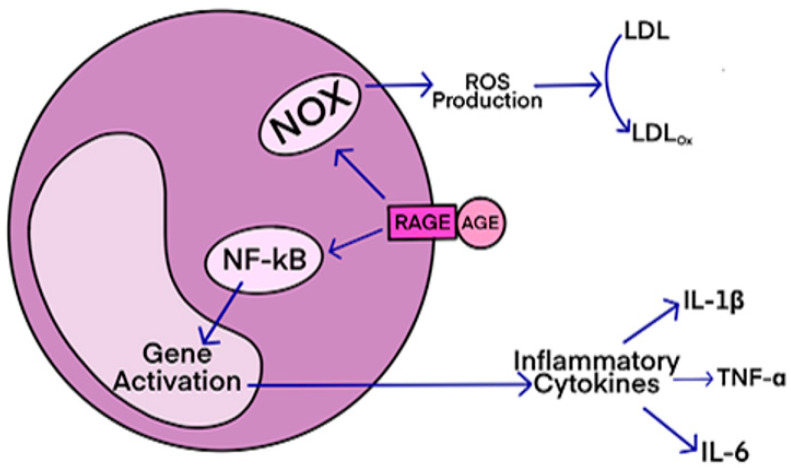
Schematic diagram showing the oxidative stress induced by AGE–RAGE binding interactions; The binding of AGEs to RAGE would lead to: (1) activation of NADPH oxidase (NOX), which translates to overproduction of ROS and thereby oxidative damage to cellular components, including low-density lipids, proteins, and nucleic acids; (2) Activation of nuclear transcription factor NF-κβ, and subsequent expression of pro-inflammatory cytokines: IL-1β, IL-6, and TNF-α.

**Figure 4 biomedicines-11-01131-f004:**
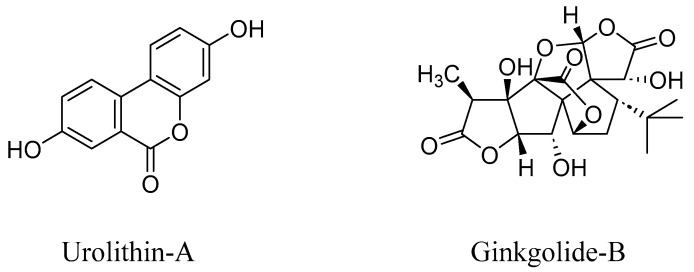
Structures of Urolithin-A, a microbiota-derived metabolite of ellaginic acid, and Ginkgolide-B, a constituent of the Ginkgo biloba leaves.

**Figure 5 biomedicines-11-01131-f005:**
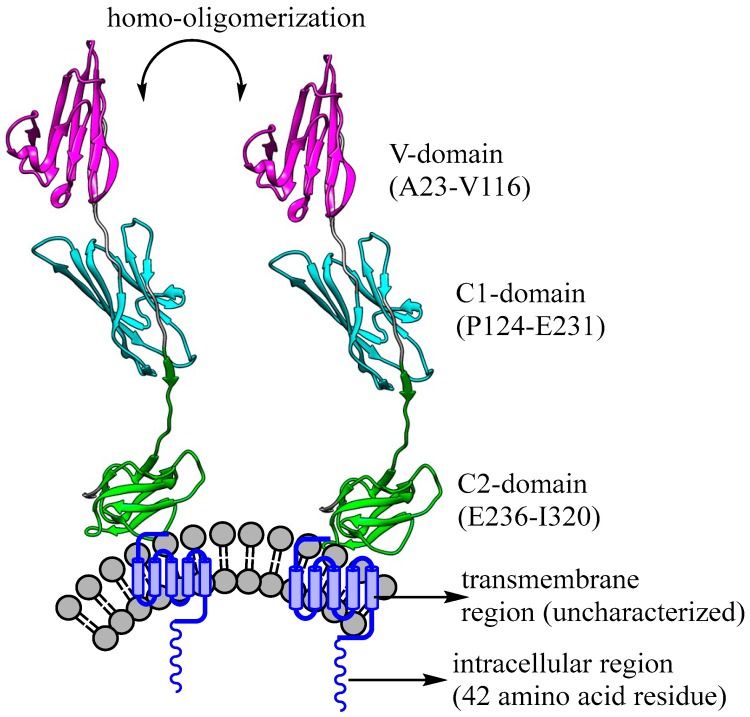
Schematic illustration of RAGE protein and its various intracellular and ectodomains; the ectodomain V-C1 (A23 to E231) is made of hydrophobic and positively charged residues, including Lys39,43,37,44,52,62,110,123,140,174,169; and Arg29,216,57,98,116,48,77,104,178, 221,228; PDB ID: 4LP5) and thereby binds to the negatively charged ligands, including AGEs, S100 family of proteins, high mobility group box1 protein (HMGB1) and β-amyloid (Aβ). The ligand binding to the VC1 domain elicits signaling cascade events transmitted through the 42-residue intracellular RAGE domain [71,72].

**Figure 6 biomedicines-11-01131-f006:**
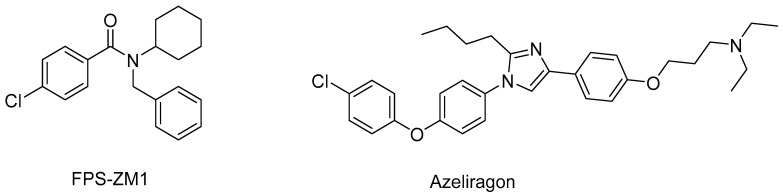
Structures of some of the RAGE inhibitors that are in clinical development.

**Figure 7 biomedicines-11-01131-f007:**
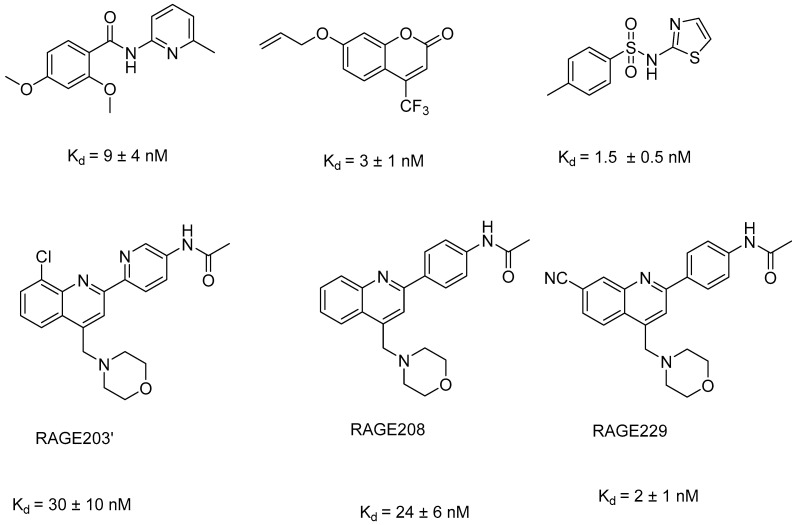
Structures of some of the lead compounds as RAGE-DIAPH1 antagonists, targeting ct-RAGE; these compounds block the extracellular AGE/RAGE mediated signaling transduction, and thereby prevent the formation of the AGE–RAGE mediated inflammatory cytokines.

## Data Availability

Not applicable.

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
