# Peer review of "RAGE Inhibitors in Neurodegenerative Diseases"

_biomedicines, 2023, doi:10.3390/biomedicines11041131_

Round 1

Reviewer 1 Report

The review entitled "RAGE Inhibitors in Neurodegenerative diseases" by Reddy P is very intersting, well done and comes to complete other reviews about RAGE and Alzheimer's disease and diagnosis and drug targetting.

But the review is missing a chapter about AGE/RAGE and inflammatory processes in neurodegenerative diseases and gut-microbiota interplay.

Conclusion, besided some misspellings, and with the addition of the chapter, the review can be published after these modifications.

Author Response

We appreciate the reviewer’s encouraging comments and suggestions and addressed all the comments as follows. These suggestions helped improve the quality of the manuscript.

The review entitled "RAGE Inhibitors in Neurodegenerative diseases" by Reddy P is very interesting, well done and comes to complete other reviews about RAGE and Alzheimer's disease and diagnosis and drug targetting.

We appreciate the reviewer for his positive and encouraging comments.

But the review is missing a chapter about AGE/RAGE and inflammatory processes in neurodegenerative diseases and gut-microbiota interplay.

We thank the reviewer for suggesting this topic, and accordingly, we have added a new section in the manuscript: “2.3. RAGE-mediated Inflammatory Processes in Neurodegenerative Diseases” (highlighted in red in the manuscript).

Conclusion, besides some misspellings, and with the addition of the chapter, the review can be published after these modifications.

We have corrected the misspellings and added a new section 2.3. on the suggested topic.  We appreciate the reviewer’s suggestions and positive comments.

Reviewer 2 Report

The article „RAGE Inhibitors in Neurodegenerative diseases” is an interesting and current review of research and molecular mechanisms on AGE-RAGE interactions and inhibition of these interactions.

The figures are well made and make it easier for the reader to understand the mechanisms described. The references are current and well selected. The article raises an important topic and can be cited many times.

minor comments:

In-text-citations should be tailored to the editorial requirements

The authors should clearly emphasize the purpose of this review

AGEs can be created endogenously, but they are also supplied in food - this should be supplemented in the text (doi: 10.1016/j.jada.2010.03.018, doi: 10.1017/S0954422419000209, doi: 10.1111/1541-4337.12593)

The authors should add information on how the database of articles was searched and how the articles were chosen

Lines 112-113 need reference

The abbreviations should be standardized - e.g. the abbreviation AD is introduced in lines 49, 306 and 325 - once introduced, the abbreviation should be consistently used in the entire text

There are many technical errors in the text - double spaces, etc.

Lines 295-303 can be removed because they are not related to the subject of the article and do not add anything to the text.

The authors recall the diseases in which there is severe glycation, but skip multiple sclerosis (doi: 10.1016/j.neuint.2013.08.009, doi: 10.3389/fimmu.2019.00855, doi: 10.15584/ejcem.2017.3.10)

Conclusions should not have citations in the text

Author Response

We appreciate the reviewer’s encouraging comments and suggestions and addressed all the comments as follows. These suggestions helped improve the quality of the manuscript.

The article „RAGE Inhibitors in Neurodegenerative diseases” is an interesting and current review of research and molecular mechanisms on AGE-RAGE interactions and inhibition of these interactions.

The figures are well made and make it easier for the reader to understand the mechanisms described. The references are current and well selected. The article raises an important topic and can be cited many times.

We appreciate the reviewer’s positive and encouraging comments, and addressed the helpful comments as follows.

minor comments:

In-text-citations should be tailored to the editorial requirements

We have now tailored the in-text-citations according to the journal format.

The authors should clearly emphasize the purpose of this review

We have now added the following para to emphasize the purpose of this review in the introduction section.

The increasing understanding of the molecular mechanisms leading to the AGE-RAGE mediated neurological disorders has broadly impacted in the design of RAGE inhibitors and antagonists for the treatment of neurological disorders and other RAGE-related diseases. Toward integrating the molecular mechanisms and recent progress in the design of RAGE inhibitors, in this perspective review we will outline the chemistry and biology of AGE-RAGE interactions and emerging small molecule based therapeutical candidates for treating neurological disorders.

AGEs can be created endogenously, but they are also supplied in food - this should be supplemented in the text (doi: 10.1016/j.jada.2010.03.018, doi: 10.1017/S0954422419000209, doi: 10.1111/1541-4337.12593)

We thank the reviewer for this suggestion and accordingly expanded the manuscript to include dAGEs and added the suggested and other relevant references (highlighted in red in the manuscript; introduction section).

The authors should add information on how the database of articles was searched and how the articles were chosen

We have added the following sentences in the introduction section to address this point.

We have used SciFinder-n® search using various key words related to AGE-RAGE interactions, AGE inhibitors and AGE antagonists, and neurological disorders.  We have emphasized on the most recent articles and also provided some of the earlier references that have fundamentally advanced the area of AGE-RAGE interactions.

Lines 112-113 need reference

We have now added references for this sentence.

The abbreviations should be standardized - e.g. the abbreviation AD is introduced in lines 49, 306 and 325 - once introduced, the abbreviation should be consistently used in the entire text

We have now consistently used abbreviations AD, TBI, and ALS throughout the manuscript.

There are many technical errors in the text - double spaces, etc.

We have now corrected the double spaces issue and other technical errors.

Lines 295-303 can be removed because they are not related to the subject of the article and do not add anything to the text.

We agree with the reviewer and accordingly we have now deleted the lines 295-303.

The authors recall the diseases in which there is severe glycation, but skip multiple sclerosis (doi: 10.1016/j.neuint.2013.08.009, doi: 10.3389/fimmu.2019.00855, doi: 10.15584/ejcem.2017.3.10)

Conclusions should not have citations in the text      

We have now deleted the citations in the conclusion section.

Round 2

Reviewer 1 Report

The authors answers all reviewers inquiries.

It can be published